# Exploring Entropy-based Active Learning for Fair Brain Segmentation

**Ghazal Danaee**[1] (iD)                                                            GHAZAL.DANAEE.1@ENS.ETSMTL.CA

**Mélanie Gaillochet**[1,2,3] (iD)                                           MELANIE.GAILLOCHET.1@ENS.ETSMTL.CA

**Christian Desrosiers**[1]                                                  CHRISTIAN.DESROSIERS@ETSMTL.CA

**Hervé Lombaert**[1,2,3] (iD)                                                    HERVE.LOMBAERT@POLYMTL.CA

**Sylvain Bouix**[1] (iD)                                                          SYLVAIN.BOUIX@ETSMTL.CA

[1] *École de technologie supérieure, Montréal, QC, Canada*

[2] *Polytechnique Montréal, Montréal, QC, Canada*

[3] *Mila - Quebec AI Institute, Montréal, QC, Canada*

**Editors:** Accepted for publication at MIDL 2026

## Abstract

Active learning (AL) has emerged as a crucial strategy for reducing the prohibitive costs associated with medical image segmentation. However, standard uncertainty-based AL methods typically focus on maximizing performance metrics, ignoring performance disparities or fairness across groups with sensitive attributes. While fair active learning has been explored in classification tasks, its intersection with medical image segmentation remains unaddressed. In this work, we introduced a fairness-aware active learning framework with a *Weighted Entropy* selection strategy that modulates uncertainty based on current group-specific performance estimates on the labeled set. To decouple true epistemic uncertainty from anatomical volume variances, we further utilized a masked, scaled entropy restricted to the region of interest. The framework was evaluated on synthetic T1-weighted brain MRIs with controlled left caudate bias in both strong and weak bias settings. A 3D U-Net was trained to segment the left caudate under several AL strategies, starting from both demographically balanced and strongly imbalanced initial labeled sets. Experiments demonstrated that our method markedly reduces performance disparities between groups compared to random sampling and standard uncertainty sampling. By prioritizing poorly segmented subgroups during the AL cycles, our method consistently achieved the highest equity-scaled performance and reduced the disparity metric by 75% (strong bias) and 86% (weak bias) relative to standard entropy at the final budget. Overall, this work is among the first studies on fair AL for medical image segmentation, offering an efficient strategy to train more equitable models in resource-constrained environments.

**Keywords:** Active learning, Brain MRI, Fairness, Segmentation.

## 1. Introduction

Active learning (AL) has become a key strategy for addressing the problem of annotation in medical image segmentation. The success of deep learning models for segmentation relies heavily on high-quality data labeled voxel by voxel by experts, which is time-consuming and labor-intensive to obtain. Active learning targets this bottleneck by treating annotation as a limited resource. The process begins with a small labeled set and a large unlabeled pool. At each iteration, a model is trained on the current labeled data, an informativeness score is computed for each unlabeled sample, and a small batch of high-scoring samples is selected for expert annotation and added to the training set (Budd et al., 2021). This loop repeats

until the labeling budget is exhausted. AL can substantially reduce annotation effort while maintaining segmentation accuracy, especially in the low-label regime (Camilleri et al., 2024).

However, medical image segmentation poses specific challenges for active learning. Unlike natural image datasets, medical image annotation cannot be easily crowdsourced; annotators must have substantial expertise, and privacy concerns further constrain data sharing (Wang et al., 2024). The task is intrinsically high-dimensional, and naive uncertainty-based sampling tends to select many highly similar, redundant images or outliers (Munjal et al., 2022). Representative-based and hybrid AL strategies mitigate this by encouraging diversity, but since they require computing distances or distributions in a learned feature space, they can be computationally expensive (Gaillochet et al., 2023). Theoretically, an advantage of active learning could be the mitigation of bias. In a simulated fraud-detection task, Weerts et al. (2023) showed that standard uncertainty-based active learning can mitigate selection bias and improve fairness even without a fairness-specific design. By querying uncertain samples, the model explored underrepresented groups, reducing false positive disparities and yielding fairer predictions as a side effect.

Beyond pure performance, a critical emerging concern is fairness. Fairness in segmentation entails ensuring that the quality of the segmentation is comparable across groups defined by sensitive attributes, including race and sex. Although most of the fairness evaluation work in medical imaging has focused on classification (Mehrabi et al., 2022), recent segmentation studies show clear demographic disparities. In cardiac MR and orthopedic imaging, models trained on racially imbalanced data exhibit significantly different performance across racial groups (Puyol-Antón et al., 2021, 2022; Lee et al., 2025; Siddiqui et al., 2024). Similar effects have been reported for prostate and skin-lesion segmentation, where race and skin-tone imbalance in training data leads to reduced performance on black patients and darker skin types (Alqarni et al., 2024; Benčević et al., 2024). In brain MRI, Ioannou et al. (2022) demonstrated that FastSurferCNN exhibits region-specific sex and race biases, noting that race-related disparities can exceed sex-related ones. Furthermore, our prior work showed that race matching between training set and test sets can substantially improve performance for some architectures but not others when segmenting the nucleus accumbens (Danaee et al., 2025). These findings build on evidence that anatomical differences across sex and racial groups shape brain volumes and model behavior (Frazier et al., 2008; Dibaji et al., 2024; Isamah et al., 2010). Fair segmentation, therefore, requires demographically balanced training cohorts, and equity-aware metrics such as Equity-Scaled Segmentation Performance (ESSP) (Tian et al., 2024).

To our knowledge, fair active learning for segmentation in a single domain remains unexplored. Wang et al. (2025) addressed a related but distinct problem: fairness in cross-domain medical image segmentation. Their method leveraged CLIP (Radford et al., 2021) to encode target-domain images and sensitive attributes. It also introduced an attribute-aware sampling strategy, coined FairAP, that enforces balanced annotation quotas across subgroups and selects representative samples in the VLM latent space.

**Our contribution**: In this work, we address group-wise fairness within the AL acquisition process for brain MRI segmentation. We introduce a novel weighted entropy strategy which modulates voxel-wise uncertainty with group-specific performance weights. These weights are derived from the current Dice score on the labeled set to prioritize samples from

under-performing groups. To ensure the acquisition score reflects true epistemic uncertainty and not anatomical variance, we compute a scaled entropy within a dilated region of interest (ROI) mask. The scaled entropy focuses on boundary uncertainty and prevents larger structures from systematically biasing the selection process across demographic groups.

## 2. Related works

**Active Learning in Segmentation.** We next summarize recent AL approaches for medical image segmentation. Atzeni et al. (2022) targeted expected dice gain per unit of manual contour length (tracing effort) for histology, and suggested selecting a specific region of interest (ROI) in one of the images for manual delineation. Kim et al. (2024) used image-level uncertainty with redundancy control for brain tumors, while Boehringer et al. (2023) prioritized the most difficult BraTS cases while pseudo-labeling the easier ones. Additionally, Qu et al. (2024), with their DifABAL, selected a compact, representative labeled core in a diffusion-learned latent space. To address the issue of redundancy in uncertainty sampling, Gaillochet et al. (2023) proposed active learning with stochastic batches. This simple but powerful add-on leverages randomness by generating batches of samples randomly and choosing the batch with the highest mean uncertainty, effectively improving diversity without complex computations.

**Fair Active Learning.** While AL for segmentation is well-studied, previous studies regarding fair active learning focus mainly on classification tasks. Anahideh et al. (2022) introduced an expected fairness metric to estimate the impact of each unlabeled sample on group-wise disparity. Their acquisition strategy prioritizes high-entropy instances while favoring those expected to reduce unfairness. Similarly, Yang et al. (2023) attempted to balance model utility and fairness by querying informative instances for both class and group labels and utilizing a sensitive learner to infer missing attributes. Wang et al. (2022) sought to improve classifier fairness by penalizing differences in true positive/false positive rates between groups and requesting more annotated samples from the worst-performing group. More recently, Fajri et al. (2024) applied fair k-means to the most uncertain points to obtain clusters that reflect the overall group distribution. They then selected candidates across clusters using a composite score that combines uncertainty and representativeness. Pang et al. (2024) aimed to mitigate unfairness among groups without compromising accuracy, using group annotations only on a validation set to preserve privacy. They achieved this by evaluating the impact of each new case on validation accuracy and fairness through an expected risk analysis. Their overall goal was to construct a fairness-aware dataset after active sampling.

## 3. Method

### 3.1. Data

To have control over the level of unfairness in the data, we generated synthetic T1-weighted brain MR images using the SimBA framework (Stanley et al., 2023). In this framework, images are derived by applying non-linear diffeomorphic transformations sampled from a learned space of deformations to a template image. The global deformations can be used to mimic "regular" anatomical variation and localized deformations can be utilized to show localized "bias or disease" effects. The deformation(s) applied to each case is unique and

controlled by sampling from a principal component (PC) representation of deformation fields. For our experiment, we combined the global transformation with an additional localized deformation in the left caudate. We denote as Group 1 the cases generated with both the localized deformation and the global deformation, and as Group 2 the cases generated with the global deformation only. The amount of localized deformation used to have a bias effect is varied by scaling the first component of the PC representation by a scalar sampled from $\mathcal{N}(\mu, \sigma)$. This procedure enabled us to construct two bias-strength conditions across Groups 1 and 2: the "strong bias" dataset with $\mu = 4$ and $\sigma = 2$ and the "weak bias" dataset with $\mu = 2$ and $\sigma = 2$. Ultimately, the weak bias dataset comprised 312 T1-weighted MRIs, including 156 cases exhibiting the bias effect and 156 cases without it. We generated a strong bias dataset of the same size (312 images), likewise balanced between biased and non-biased cases (156 each). All images had the resolution of $170 \times 170 \times 76$ voxels with an isotropic voxel spacing of 1mm.

### 3.2. Weighted localized entropy

We introduce two main modifications to the naive use of entropy. First, we limit the computation of entropy within a mask around the ROI. For each unlabeled candidate volume, let $\mathcal{R}$ denote the ROI and $H(v)$ the voxel-wise predictive entropy at voxel $v$. We define a masked, scaled entropy:

$$\hat{H} = \frac{1}{|\mathcal{R}|} \sum_{v \in \mathcal{R}} H(v), \tag{1}$$

which averages uncertainty over the ROI while normalizing by the region size. This reduces the influence of trivial anatomical volume differences on the overall uncertainty score. In our case, $\mathcal{R}$ is a dilated mask around the predicted segmentation of the left caudate.

Second, a group-aware weighting scheme that re-weights the scaled entropy with group-specific weights, thereby prioritizing samples from groups on which the model underperforms. For each group $g$, we first compute a standardized performance score based on the Dice similarity coefficient (DSC) on the labeled set: $z_g = \frac{\overline{\mathrm{DSC}}_{\mathrm{all}} - \overline{\mathrm{DSC}}_g}{\sigma_{\mathrm{all}}}$, where $\overline{\mathrm{DSC}}_g$ denotes the mean dice for group $g$, $\overline{\mathrm{DSC}}_{\mathrm{all}}$ is the mean dice computed over all labeled set, and $\sigma_{\mathrm{all}}$ is the corresponding standard deviation across the labeled set. Thus, groups with worse segmentation performance (lower $\overline{\mathrm{DSC}}_g$) yield larger $z_g$. We then transform these standardized scores into normalized group weights via a softmax: $w_g = \frac{\exp(z_g)}{\sum_j \exp(z_j)}$. The final acquisition score used for selection is then given by

$$\mathrm{score}(x_g) = w_g \cdot \hat{H}, \tag{2}$$

so that uncertainty is explicitly re-weighted toward groups with relatively poorer DSC, ensuring that active learning focuses on groups where the model is currently less reliable.

## 4. Experiment and Results

### 4.1. Implementation details

**Model architecture and training setup.** We have summarized detailed information about the network, active learning setup, and test data in Table 1.

Table 1: Summary of the network training setup, test data, and the active-learning configuration. Group 1 denotes cases with an additional localized deformation in the left caudate, while Group 2 contains only global deformation.

| Network | Test data | Active learning |
|---|---|---|
| **3D U-Net** (GroupNorm, 3D convolution, and ReLU blocks, with a sigmoid activation, 2-level configuration with feature maps of size 8, 16) **Optimizer:** Adam **Epochs:** 200 **Learning rate:** $10^{-4}$ **Loss:** binary cross-entropy | **Test sets:** **Group 1:** $N = 30$ **Group 2:** $N = 30$ **Combined (Group 1 ∪ Group 2):** $N = 30$ (15 from Group 1, 15 from Group 2) | **Strategies:** • Random sampling • Mean entropy sampling • Localized entropy sampling • Weighted localized entropy sampling **AL schedule:** 5 full AL cycles (10–86 labeled) **Batch size:** $b = 4$ |

**Baseline Experiments.** We first establish baseline fairness by training our model under three training set compositions. The composition details are summarized in Table 3. For both the strong and weak bias datasets, the test set comprised 30 cases, including 15 cases from Group 1 and 15 cases from Group 2.

**Active Learning.** In all AL experiments, we start with 10 labeled data, select 4 new samples to label at each AL iteration, and perform five complete AL cycles. We investigate three scenarios with varying proportions of Group 1 and Group 2 images in the initial training set. The corresponding compositions are reported in Table 2.

Table 2: Training data configuration by bias strength. Group 1 denotes cases with an additional localized deformation in the left caudate, while Group 2 contains only global deformation.

| Bias strength | Training data |
|---|---|
| **Strong bias** | **Initial labeled set:** $N_0 = 10$ **Initial group proportions (Group 1/Group 2):** • 50/50 • 80/20 • 20/80 |
| **Weak bias** | **Initial labeled set:** $N_0 = 10$ **Initial group proportions (Group 1/Group 2):** • 50/50 • 80/20 • 20/80 |

We implemented and tested four different AL strategies on the same test set used in the baseline experiments. We compared our fairness-weighted localized entropy sampling with random sampling (RS), mean entropy sampling, and localized entropy sampling (eq.(1)). We report DSC, ESSP, and $\Delta$ for each experiment.

## 4.2. Evaluation metrics

We used DSC to evaluate raw performance. Furthermore, to evaluate fairness in the model's results, we utilized the Equity-Scaled Segmentation Performance (ESSP) metric, originally proposed by Tian et al. (2024). Given $\text{DSC}_{overall}$, the average DSC over all cases, and $\text{DSC}_g$, the average DSC of group $g$, we first define $\Delta$ as the sum of absolute performance discrepancies across all groups:

$$\Delta = \sum_{g \in G} \left| \text{DSC}_{overall} - \text{DSC}_g \right|. \tag{3}$$

ESSP is then computed by penalizing the overall performance with $\Delta$:

$$\text{ESSP} = \frac{\text{DSC}_{overall}}{1 + \Delta}. \tag{4}$$

In essence, ESSP acts as a substitute for DSC, with a penalty for unfairness.

## 4.3. Baseline experiments

In the baseline experiments, we use all the training data available to get baseline Dice score (DSC), ESSP and $\Delta$. Results are shown in Table 3. Surprisingly, the balanced dataset does not lead to the best ESSP. Instead, a model trained exclusively on Group 1 (the deformed dataset) leads to better ESSP than the other scenarios. The baseline experiments characterize how different training cohort compositions (balanced, Group 1-only, Group 2-only) affect overall accuracy. They also quantify equity using ESSP and $\Delta$ in the absence of active selection. This provides a reference point to assess whether subsequent AL strategies offer fairness gains beyond what simple cohort design can achieve.

## 4.4. Active learning experiments

Results for the experiments starting from a balanced dataset (5 samples from each group) are shown in the first row of Fig. 1 (ESSP). The 80/20 and 20/80 Group 1/Group 2 initializations are shown in rows 2 and 3, respectively. All methods start from the same training set and thus exhibit identical performance at the first cycle (10 labeled). We also show $\Delta$ and DSC curves for the strong bias experiment (Fig. 2). Results are very similar for the weak bias experiment and can be found in the appendix. Overall, weighted localized entropy outperforms all methods in terms of ESSP in all scenarios, followed closely by localized entropy, then random sampling. Global entropy performs worse than all methods by a relatively large margin. In terms of raw performance as measured by DSC (Fig. 2), random sampling is often the best strategy, especially at the early stages of AL. However, this naive strategy fails to perform as well in reducing $\Delta$ effectively, compared to localized entropy and the proposed weighted localized entropy (Fig. 2). The significant lowering of $\Delta$ by weighted localized entropy, while still remaining highly competitive in terms of DSC, allows it to achieve the top ESSP scores across most experiments.

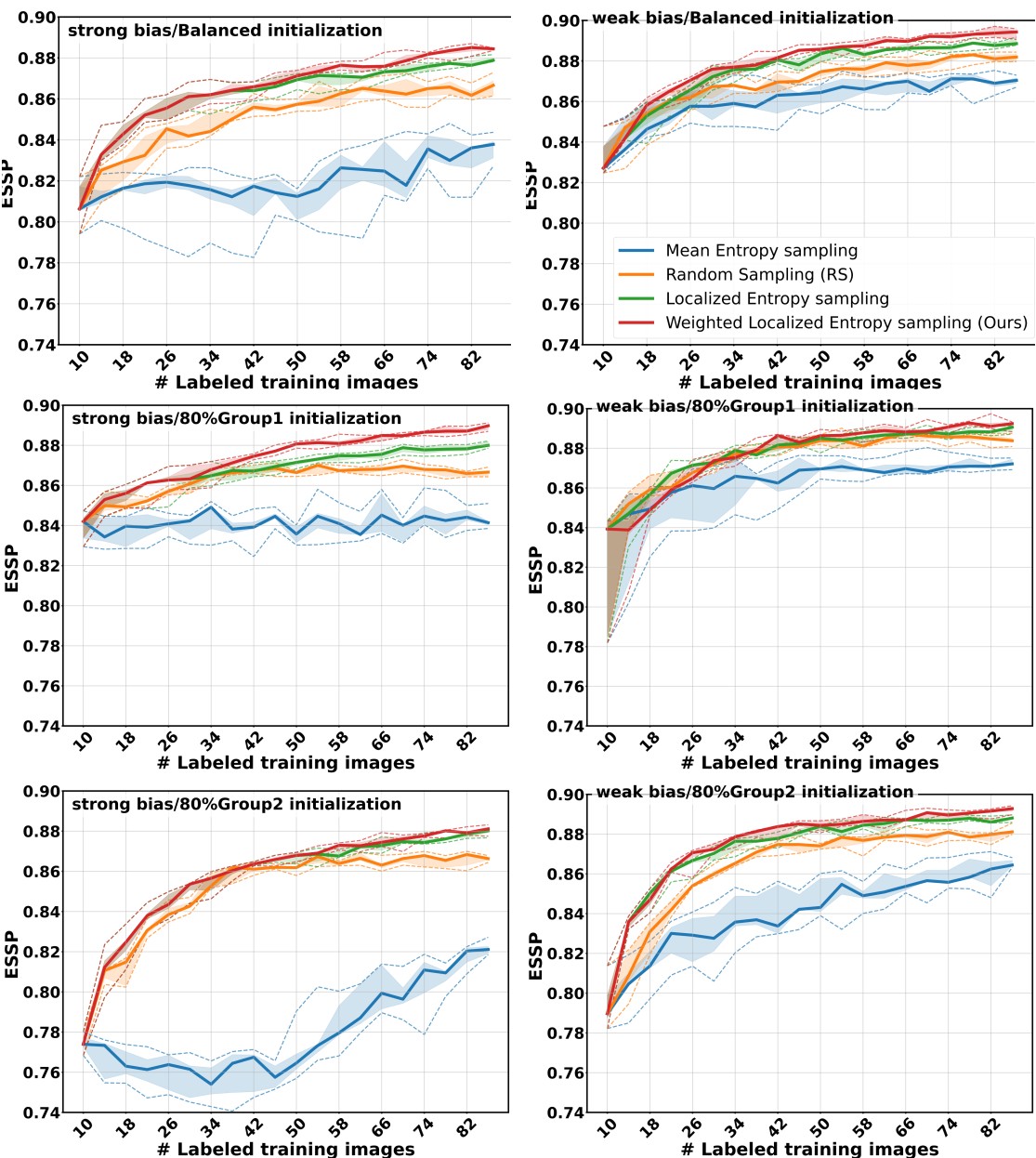

Figure 1: **ESSP** under different initial training set compositions. First row: balanced initialization, second row: 80/20 Group 1/Group 2 ratio, third row: 20/80 ratio. Left column: strong bias dataset, Right column: weak bias dataset.

Table 3: Segmentation performance (DSC) stratified by training cohort and evaluated on Group 1, Group 2, and pooled test sets of the strong bias and weak bias datasets. Group 1 denotes cases with an additional localized deformation in the left caudate, while Group 2 contains only global deformation. The size of the training set is written in parentheses.

| Training | Test | | | | |
|---|---|---|---|---|---|
| | DSC(G1) | DSC(G2) | DSC(G1∪G2) | ESSP | Δ |
| **Strong Bias** | | | | | |
| Pooled ( 63 G1 + 63 G2 ) | 0.88 | 0.93 | 0.91 | 0.91 | 0.04 |
| Group 1 ( 126 ) | 0.89 | 0.88 | 0.89 | 0.89 | 0.01 |
| Group 2 ( 126 ) | 0.75 | 0.93 | 0.84 | 0.71 | 0.18 |
| **Weak Bias** | | | | | |
| Pooled ( 63 G1 + 63 G2 ) | 0.90 | 0.91 | 0.90 | 0.89 | 0.01 |
| Group 1( 126 ) | 0.90 | 0.90 | 0.90 | 0.90 | 0.002 |
| Group 2 ( 126 ) | 0.86 | 0.91 | 0.88 | 0.84 | 0.04 |

**Selection dynamics and group composition.** As demonstrated in the baseline experiments (Table 3), the best ESSP is likely achieved by over-representing Group 1 in the training dataset. This is illustrated in Fig. 3, where, under all scenarios, the weighted localized entropy consistently favors adding Group 1 samples to the training dataset. One can also observe a link between localized entropy and fairness as this strategy also tends to select samples from Group 1, even though it does not explicitly account for fairness. Random sampling behaves as expected, balancing data 50/50 over time, while global entropy behaves counterintuitively by adding more samples from Group 2.

## 5. Discussion

In this work, we investigated the intersection of active learning and fairness in medical image segmentation. We designed an acquisition algorithm to improve group-wise fairness rather than solely optimizing accuracy. The proposed Weighted Localized Entropy consistently achieved the strongest equity-aware performance across initialization regimes and bias strengths. Unsurprisingly, localized entropy also reduced bias, although applying the group-fairness weights further yielded improvement in both fairness and accuracy. We note that random sampling usually acquired the best accuracy results, but the equity-scaled performance was harmed by the substantial group-wise performance disparity. Global entropy frequently became the worst-performing method in terms of accuracy and fairness as it tended to add more Group 2 cases over time, despite Group 1 being the morphology-challenged subgroup. This behavior can be because whole-volume uncertainty may be dominated by structural extent and incidental variability rather than meaningful model confusion at the target boundary. By computing entropy inside an ROI mask and normalizing by the ROI size (Eq. 1), the localized entropy better isolates the uncertainty.

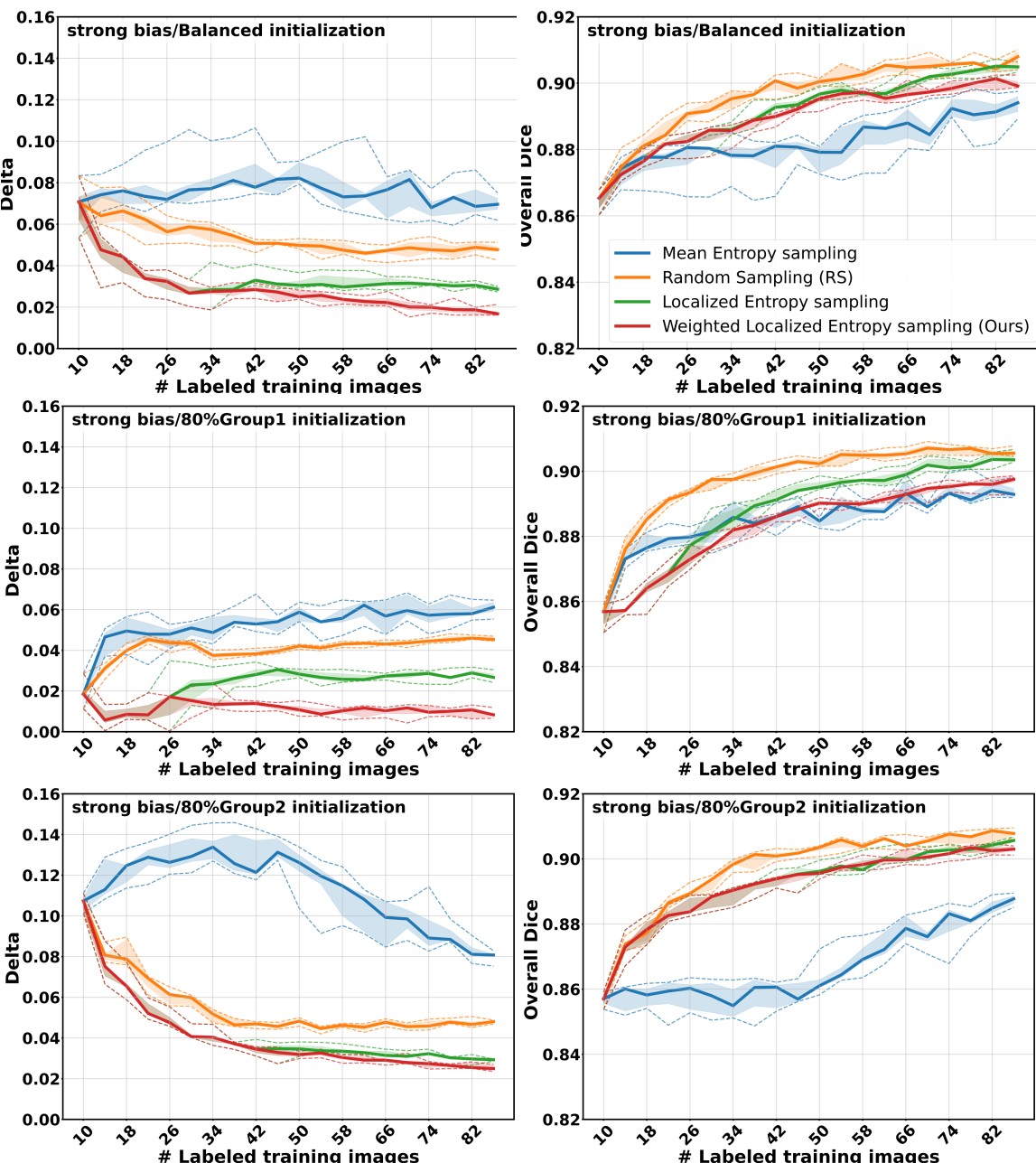

Figure 2: **Δ** and **DSC** metrics under different initial training set compositions for the strong bias experiment only. First row: balanced initialization, second row: 80/20 Group 1/Group 2 ratio, third row: 20/80 ratio. Left column: Δ, Right column: DSC.

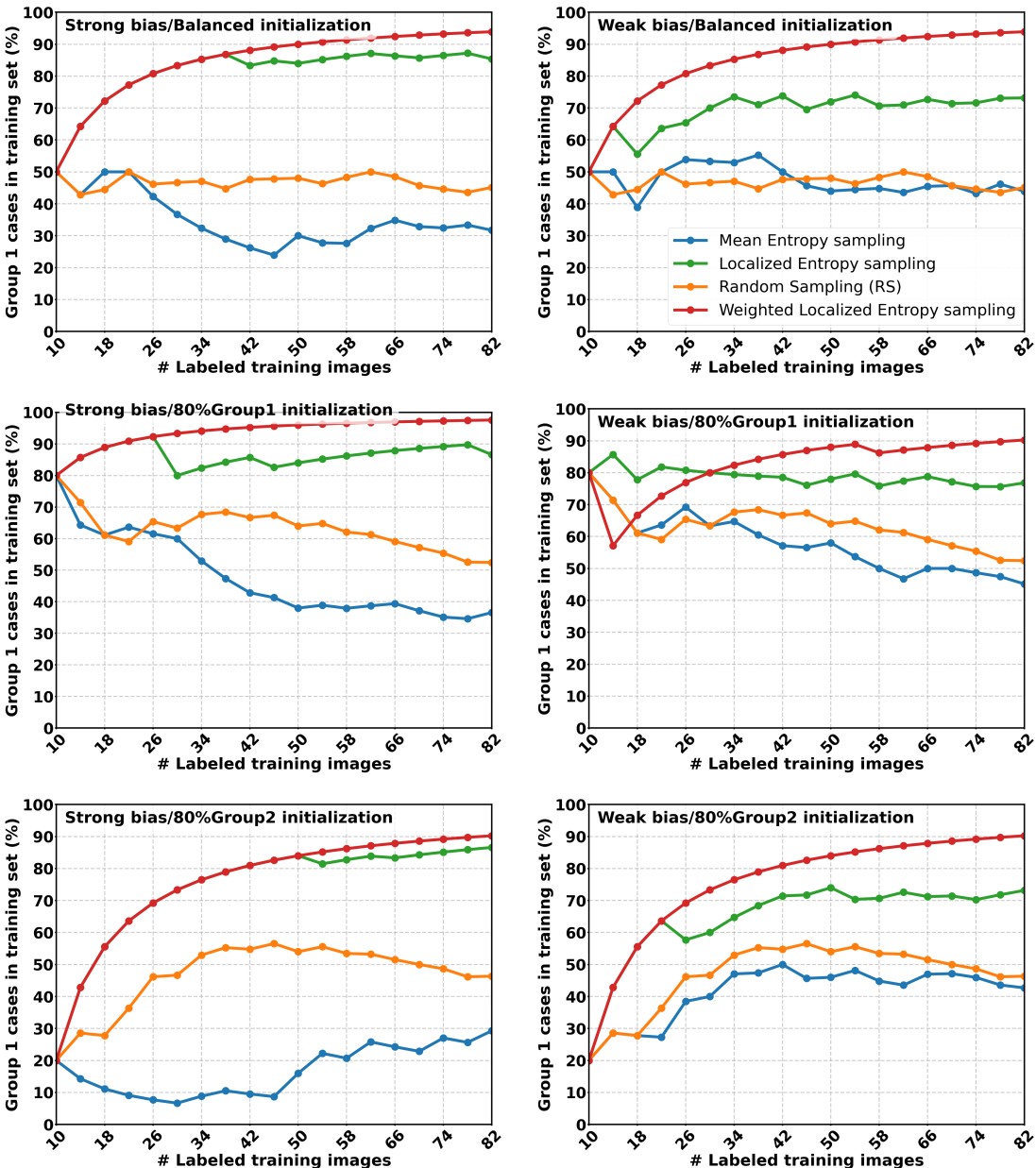

Figure 3: **Group 1 ratio** in the training set after sampling for each cycle under different initial training set. First row: balanced initialization, second row: 80/20 Group 1/Group 2 ratio, third row: 20/80 ratio. Left column: strong bias dataset, Right column: weak bias dataset.

The proposed method demonstrated robustness in both strong and weak bias scenarios. In the strong bias setting, we observed a reduction in $\Delta$ of approximately 75% (0.0176 vs. 0.0692) relative to standard entropy at the final cycle. Notably, in the weak bias setting in which morphological differences are harder to detect, our method was even more effective relative to baselines, reducing disparity by approximately 86%. This indicates that the weighted entropy signal is sensitive enough to detect and correct minor performance drifts.

As it was observed in Table 3, the relationship between training composition and equitable performance is non-trivial. In the weak bias dataset, the Group 1-only configuration became the best fairness baseline (DSC=0.90, ESSP=0.90, $\Delta = 0.002$), outperforming training on all groups (DSC=0.90, ESSP=0.89, $\Delta$=0.01). Overall, these outcomes suggest that the subgroup associated with more challenging morphology (Group 1) acts as a fairness anchor and overrepresenting it can reduce group disparity without severely compromising overall utility. This observation motivates the core design of our AL strategy, which adaptively increases the selection pressure toward the currently under-performing group.

We tackle the AL for segmentation with a lightweight mechanism that uses only labeled-set performance to estimate group weights and requires no additional fairness classifier or expensive representativeness modeling. Our approach is close to performance-driven group reweighting strategies proposed in classification (Wang et al., 2022), but adapted to voxelwise uncertainty.

Although one might suspect overfitting to Group 1, this is not supported in the classical sense: when trained only on Group 1, the model still generalizes well to Group 2, with Group 2 performance remaining close to Group 1. Additionally, the performance differences when predicting Group 2 across all baseline experiments are only minor.

We argue that the higher ESSP achieved by training on Group 1 only is not driven by overfitting, but by robust generalization from training with the more morphologically challenging Group 1 dataset. Group 1 includes both global inter-subject variability and additional localized deformations. A model trained on Group 1 learns features that remain valid when evaluated on Group 2, where the task is effectively easier because the localized deformations are absent. In contrast, training on pooled Group 1 and Group 2 data can encourage shortcut learning, where the model preferentially fits the easiest, most frequent patterns (Group 2) and underfits the complex Group 1 cases, consistent with the gaps observed in Table 3.

Importantly, Group 1-only training is not presented as a universally optimal deployment strategy. ESSP is not a pure accuracy metric: it explicitly penalizes between-group disparity through ($\Delta$), defined as the sum of absolute deviations from the overall dice. Since Group 1 is the morphology-challenged subgroup by construction, the fairness gain under Group 1-only training increases. Group 2 performance remains high while the inter-group gap shrinks, and this is exactly what ESSP rewards. Moreover, the Table 1 results are computed on a held-out, balanced test set (15 subjects per group), so the effect is not memorization-based overfitting.

We agree that one limitation of this study is the use of a synthetic dataset, which we selected to explicitly control the presence and magnitude of morphological bias. This controlled setting enables a clear link between sampling strategy and performance bias. Real-world medical data contains complex biases (e.g., scanner artifacts correlated with hospital demographics) that are harder to disentangle than anatomical deformations. *Eval-*

*uating* our framework in real-world data scenarios requires (i) identification of a dataset with known biases to confirm that a measurable disparity exists, and (ii) reliable reference segmentations for training, ideally generated without performance biases. In practice, finding such datasets is extremely challenging. Automatic segmentations may carry performance biases making fairness evaluation challenging, and manual annotations by experts are scarce. Moreover, real cohorts may exhibit subtle or region-specific morphometric differences (or none at all), and the existence of such differences cannot be assumed a priori. These challenges led us to perform these experiments solely with synthetic data. Our results support the use of a weighted sampling strategy to avoid performance bias associated with group level attributes. If one were to *translate* or apply our framework in a real-world scenario, one would need to identify one or more sensitive attributes based on apriori hypotheses (structure X has been reported to be larger in sub-population Y) and guide the sampling strategy using the weighted localized entropy (eq. 2 2).

Another limitation of our work is that our weighting strategy relies on the availability of group labels for the labeled set. While it is a reasonable assumption in a controlled AL setup, extending this to scenarios where sensitive attributes are missing is a necessary future step.

## 6. Conclusion

We presented a fairness-aware active learning framework for brain MRI segmentation. Through using a performance-based weighting scheme and localized entropy, the proposed algorithm actively constructs a training set that prioritizes equity. This can be especially practical for deploying segmentation models in settings where both labeling budgets and fairness requirements are critical. Our study provides a robust foundation for advancing fair active-learning approaches in medical image segmentation.

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

## Appendix A. Δ and DSC results for the weak bias experiments

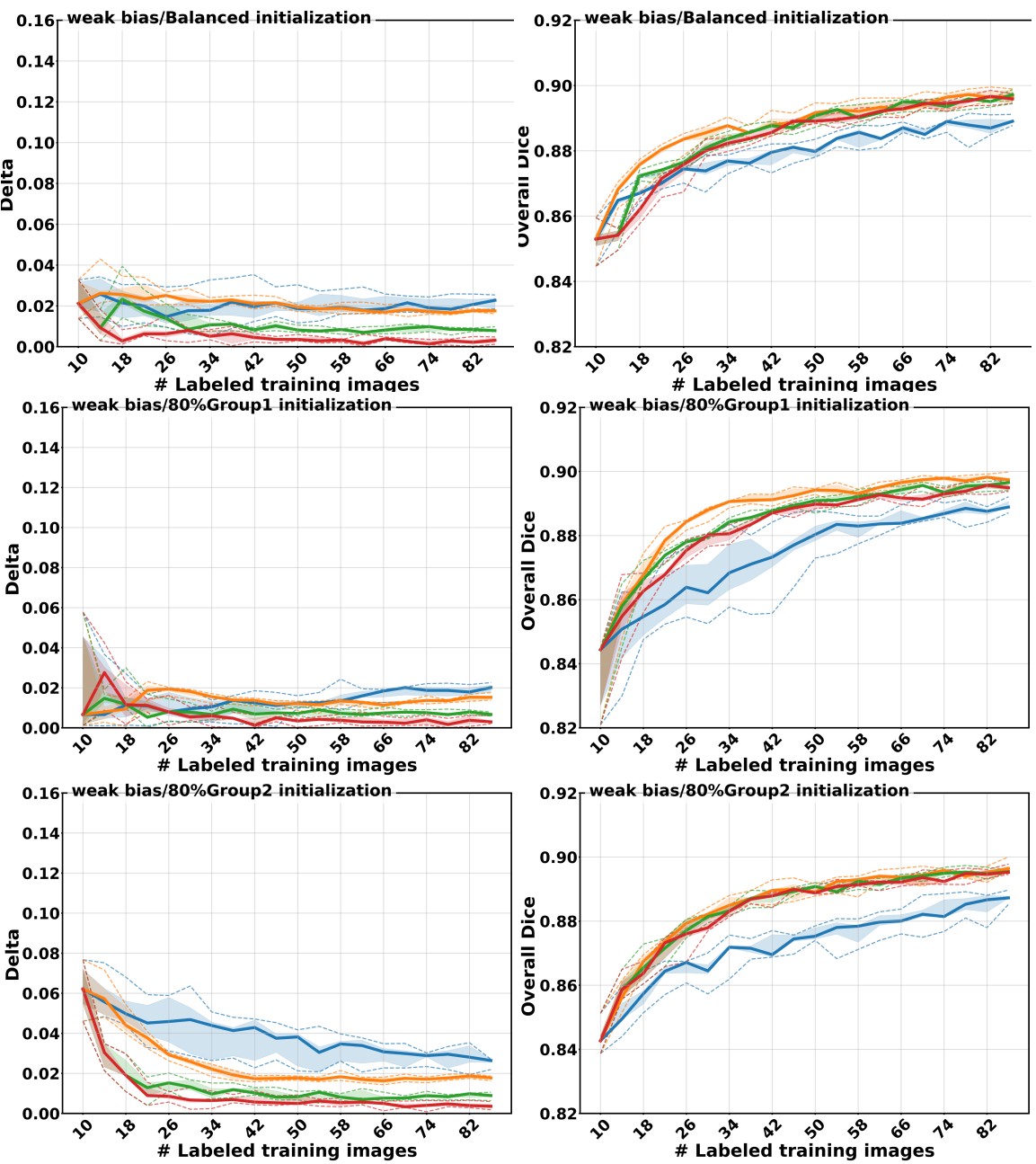

Figure 4: Performance metrics under different initial training set compositions for the weak bias experiments. First row: balanced initialization, second row: 80/20 Group 1/Group 2 ratio, third row: 20/80 ratio. Left column: Δ, Right column: DSC.

