# OpenReview forum: "Exploring Entropy-based Active Learning for Fair Brain Segmentation"
_MIDL.io/2026/Conference — MIDL 2026 Poster_

### Official Review · Reviewer_qMDW · 2025-12-29

**Confidence:** 3
**Preliminary Rating:** 2
**Final Rating:** 3

**Summary:**

The authors propose a fairness-aware active learning (AL) strategy for medical image segmentation, introducing Weighted Localized Entropy, which modulates voxel-wise uncertainty by group-specific performance gaps. The method is evaluated on synthetic T1-weighted brain MRI scans with controlled morphological bias in the left caudate. Results show substantial reductions in group-wise disparity (∆) and improvements in equity-scaled segmentation performance (ESSP), relative to standard entropy and random sampling.

**Strengths:**

The problem tackled in this manuscript is timely and its motivation is well grounded. The authors address a clear gap in the current literature: fair active learning for medical imaging segmentation. The weighted entropy formulation is elegant and seems pretty straightforward to implement. The results section shows consistent empirical gains with the proposed method compared to random sampling, while keeping a competitive dice coefficient.

**Weaknesses:**

The use of purely synthetic data is an evident limitation of this work. This is a good proof of concept which, however, does not show that this method would apply also to real world clinical datasets where the bias is multi-factorial (scanner, site, protocol) and often correlated with random variables. Furher, the authors focus on a single structure (caudate), limiting the scope of the segmentation task.

**Detailed Comments:**

-Please clarify how often group weights are recomputed (every AL cycle?).

-The authors assume simple and known group labels, but in practice group membership is often missing, sensitive, or noisy. The authors should discuss how this method might extend to latent or proxy group labels.

-Figure 1 and 2 should be simplified. Most of these plots could be in the appendix.

**Justification Of Final Rating:**

The authors have addressed most of my comments in the rebuttal and I have increased my rating. However, the proposed method still lacks a validation on a real dataset which would make it much stronger.

**Justification Of The Preliminary Rating:**

The authors tackle an extremely important problem in medical imaging segmentation: fairness across different groups. The proposed method, however, it's tested only on a single segmentation task from synthetic data and doubts remain about its applicability on real datasets.

**Questions To Address In The Rebuttal:**

-Can the authors reproduce the results on a small real dataset (e.g., ADNI, UK Biobank subset, or other public available dataset)? It would be great to verify the propose method when the bias is noisier and group labels are different.

-From the results it seems that training only on the biased (Group 1) data yields better ESSP than training on all data. This is an interesting finding which raises several questions. Is fairness being improved because Group 2 is particularly easy? Or does ESSP reward overfitting to the hardest group? This point deserves a much more detailed discussion.

---

> ### Author Response · Authors · 2026-01-25
>
> **(Q1)** Can the authors reproduce the results on a small real dataset (e.g., ADNI, UK Biobank subset, or other public available dataset)? It would be great to verify the propose method when the bias is noisier and group labels are different?
>
> **Response:** We thank the reviewer for these important comments. When we initially set out to do this project we were aiming to perform all experiments with data from open data projects (e.g. openneuro). Unfortunately, as described in response to reviewer BUYe comment, finding datasets with known morphological differences between subpopulation and unbiased segmentation for training and testing is extremely challenging. We therefore decided to solely concentrate on a synthetic dataset for which we have full control of the level of morphological differences and have a true ground truth labeled data.
>
>
> **(Q2)** From the results it seems that training only on the biased (Group 1) data yields better ESSP than training on all data. This is an interesting finding which raises several questions. Is fairness being improved because Group 2 is particularly easy? Or does ESSP reward overfitting to the hardest group? This point deserves a much more detailed discussion.
>
> **Response:** We agree this is an important baseline observation, and we will expand the discussion. Although one might suspect overfitting to Group1, this is not supported in the classical sense: when trained only on Group1, the model still generalizes well to Group2, with Group2 performance remaining close to Group1.
>    Additionally, the performance differences when predicting Group2 across all baseline experiments are only minor.
>
>    We argue that the higher ESSP achieved by training on Group1 only is not driven by overfitting, but by robust generalization from training with the more morphologically challenging Group1 dataset. Group1 includes both global inter-subject variability and additional localized deformations. A model trained on Group1 learns features that remain valid when evaluated on Group2, where the task is effectively easier because the localized deformations are absent. In contrast, training on pooled Group1 and Group2 data can encourage shortcut learning, where the model preferentially fits the easiest, most frequent patterns (Group2) and underfits the complex Group1 cases, consistent with the gaps observed in Table1.
>
> Importantly, ESSP is not a pure accuracy metric: it explicitly penalizes between-group disparity through $(\Delta)$, defined as the sum of absolute deviations from the overall dice. Since Group1 is the morphology-challenged subgroup by construction, the fairness gain under Group1-only training increases. Group2 performance remains high while the inter-group gap shrinks, and this is exactly what ESSP rewards. Moreover, the Table1 results are computed on a held-out, balanced test set (15 subjects per group), so the effect is not memorization-based overfitting. We will revise the Discussion to state this explicitly and to clarify that Group1-only training is not presented as a universally optimal deployment strategy.

---

> > ### Comment · Reviewer_qMDW · 2026-01-29
> >
> > Thank you for addressing most of my concerns. Validating this framework on a real dataset is still a crucial point in my opinion. While I understand the motivation for using synthetic data to enable controlled bias injection and causal analysis, a complementary experiment on a real-world dataset, even with imperfect annotations (or consensus annotations), would substantially strengthen the practical relevance of the proposed method.

---

### Official Review · Reviewer_BUYe · 2026-01-07

**Confidence:** 4
**Preliminary Rating:** 3
**Final Rating:** 4

**Summary:**

The paper tackles the challenge of fairness in medical image segmentation, and particularly in the context of active learning. While standard selection strategies leverage uncertainty, the authors propose to add two components: i) a localized entropy to take into account volume variability, and ii) a weighted computation to increase the selection of worst-group samples at next rounds of active learning. Evaluation is performed on synthetic T1-weighted brain MRIs generated with controlled global and localized deformations (localized left caudate deformation defines Group 1; global-only defines Group 2), under strong- and weak-bias regimes. The method is compared to random sampling, standard (global) entropy, and localized entropy across three initialization ratios (balanced, 80/20, 20/80). Results show that weighted localized entropy consistently improves fairness metrics (ESSP) and reduces group disparity (Δ) compared with entropy baselines, though overall Dice is often best with random sampling, indicating a fairness–accuracy trade-off. The paper reports large relative reductions in disparity versus standard entropy at the final budget (on the order of ~75%/86% in strong/weak bias settings, as stated by the authors).

**Strengths:**

* The paper addresses a timely and important challenge in fairness in medical image segmentation and, at the same time, incorporates such changes to active learning to address the issue of limited labeled data.
* The idea of optimizing the worst-case scenario and masking entropy is not novel, but it has not yet been applied to the selection process in active learning, making it a strong and interesting contribution.
* The authors clearly and thoroughly position their paper relative to state-of-the-art methods for fairness and active learning.
* The methodology is well-detailed and easy to understand. The evaluation design is well-motivated: controlled synthetic biases (strong/weak; different initial group ratios) help isolate mechanisms and show robustness across several regimes.

**Weaknesses:**

1) Organization and clarity of experimental design
* Group definitions are easy to lose: Groups 1/2 are defined in the Data/Method section, but then heavily referenced later without reminders, making the experimental narrative harder to follow. A different, more self-explanatory labelling of groups, such as local-deformation group and global-deformation group, would help.
* Baseline cohorts (balanced vs G1-only vs G2-only) and active learning initializations (5/5, 8/2, 2/8) are all described in text. A single table/diagram summarizing: dataset type (strong/weak), initialization, acquisition method, and labeling budget would significantly improve readability.
* Some referencing/labeling is confusing: e.g., the baseline section refers to “Table 4.3” while the table appears as Table 1.

2) Figures and table presentation issues
* Figures are hard to analyze and interpret quickly: many plots encode scenarios/metrics in rows/columns, but identifying each scenario relies heavily on the caption, and readers must constantly map back and forth. Consider adding small in-panel labels (e.g., “Strong bias / Balanced initialization”) and harmonizing legend placement.
* Inconsistent y-axis scales across plots for the same metric make visual comparison difficult, using consistent ranges would help.
* The “% Group 1 selected over cycles” figure is informative, but many panels repeat the same qualitative trend. Consider consolidating (e.g., show mean ± variability, or fewer representative panels).
* Table 1 is ambiguous: it is not immediately clear that “All” corresponds to the balanced cohort using both groups (as described in the baseline setup). Renaming to “Balanced” (and ensuring the column order is unambiguous) would fix this quickly.

3) Methodological limitations that should be stated more explicitly
* Assumption about group labels for unlabeled data: the acquisition score is group-weighted, which typically requires knowing (or inferring) group membership for candidates in the unlabeled pool. The paper notes reliance on group labels as a limitation, but this assumption is central and should be emphasized earlier, along with discussion of what happens when attributes are missing/noisy.
* ROI mask dependency on model predictions: localized entropy uses a dilated ROI around the predicted segmentation. Early-cycle predictions may be poor, which could bias the uncertainty measurement. It would help to clarify how the mask is produced (thresholding, dilation size) and whether results are sensitive to these choices.
* Incremental benefit vs localized entropy: localized entropy alone already improves fairness (and appears close to the proposed method in some plots). The paper would benefit from a sharper quantification of the additional gain from group-weighting (e.g., averaged final-cycle improvements with statistical testing, or an ablation that isolates weighting vs masking effects).

**Detailed Comments:**

See weaknesses.

**Justification Of Final Rating:**

I would like to thank the authors for addressing my concerns. The clarity of the experiments has been significantly improved in the revised version, and the figures are easier to read. While I concur with Reviewer qMDW on the necessity of a real-world experiment, even with weak labels, I do believe that the methodological contribution to active learning with fairness constraints should clearly be presented and therefore recommand weak acceptance.

**Justification Of The Preliminary Rating:**

The paper tackles an important challenge for fair active learning for segmentation by introducing a weighted, ROI-localized entropy score and demonstrates consistent improvements in fairness metrics across several controlled settings. However, while fairness gains are convincing, the work shows a clear fairness-accuracy trade-off (Dice is often best with random sampling), and the evaluation is restricted to synthetic deformations, leaving uncertainty about the impact on real-world bias factors. Main concerns are presentation/organization (hard-to-parse figures and experiment descriptions), and the limited realism/generalizability of the synthetic-only setup. Overall, I view the contribution as solid and relevant, with clear fairness gains, but with multiple limitations and clarity issues.

**Questions To Address In The Rebuttal:**

* Add a table summarizing all experiment conditions (weak/strong bias, group proportion).
* Improve figure readability with consistent y-axes, clear in-panel labels, and reduced redundancy.
* Clarify Table 1 labels/columns; rename “All” and ensure the table matches the baseline description.
* Expand discussion of real-world feasibility.

---

> ### Author Response · Authors · 2026-01-25
>
> **(Q1)** Add a table summarizing all experiment conditions (weak/strong bias, group proportion).
>
> **Response:** We thank the reviewer for these important comments to help clarify the manuscript. We include two tables (Table1 and Table2), which will be added to the manuscript.
>
>  **Table 1.** Summary of the network training setup, test data, and the active-learning configuration. Group 1 denotes cases with an additional localized deformation in the left caudate, while Group 2 contains only global deformation.
>
>  | Network | Test data | Active learning |
>  |---|---|---|
>  | 3D U-Net; Optimizer: Adam; Epochs: 200; Learning rate: 1e-4; Loss: binary cross-entropy | Test sets: Group 1 (N=30), Group 2 (N=30), Combined (N=30; 15 from Group 1, 15 from Group 2) | Strategies: Random; Mean entropy; Localized entropy; Weighted localized entropy; AL schedule: 5 cycles (10–86 labeled); Batch size: b=4 |
>
>
>  **Table 2.** Training data configuration by bias strength. Group 1 denotes cases with an additional localized deformation in the left caudate, while Group 2 contains only global deformation.
>
>  | Bias strength | Training data |
>  |---|---|
>  | Strong bias | Initial labeled set: 10; Initial group proportions (Group 1/Group 2): 50/50, 80/20, 20/80 |
>  | Weak bias | Initial labeled set: 10; Initial group proportions (Group 1/Group 2): 50/50, 80/20, 20/80 |
>
>
> **(Q2)** Improve figure readability with consistent y-axes, clear in-panel labels, and reduced redundancy .
>
> **Response:**
> We will change Figure1, Figure2, Figure3, and Figure4 in the revised version.
>
> **(Q3)** Clarify Table 1 labels/columns; rename “All” and ensure the table matches the baseline description.
>
> **Response:**
> We have revised the table based on the reviewer’s comment, which we agree, will help clarify the baseline experiments (Table 3).
>
>  **Table 3.** Segmentation performance (DSC) stratified by training cohort and
>  evaluated on Group 1, Group 2, and pooled test sets of the strong bias and weak
>  bias datasets. Group 1 denotes cases with an additional localized deformation
>  in the left caudate, while Group 2 contains only global deformation.
>
>  | Training | &nbsp;&nbsp;&nbsp;&nbsp;&nbsp;DSC(G1) | &nbsp;&nbsp;&nbsp;&nbsp;&nbsp;DSC(G2) | &nbsp;&nbsp;&nbsp;&nbsp;DSC(G1 U G2) | &nbsp;&nbsp;&nbsp;&nbsp;&nbsp;ESSP | &nbsp;&nbsp;&nbsp;&nbsp;&nbsp;Δ |
>  |---|---:|---:|---:|---:|---:|
>  | **Strong Bias** |  |  |  |  |  |
>  | Pooled (63 G1 + 63 G2) | 0.88 | 0.93 | 0.91 | 0.91 | 0.04 |
>  | Group 1 (126) | 0.89 | 0.88 | 0.89 | 0.89 | 0.01 |
>  | Group 2 (126) | 0.75 | 0.93 | 0.84 | 0.71 | 0.18 |
>  | **Weak Bias** |  |  |  |  |  |
>  | Pooled (63 G1 + 63 G2) | 0.90 | 0.91 | 0.90 | 0.89 | 0.01 |
>  | Group 1 (126) | 0.90 | 0.90 | 0.90 | 0.90 | 0.002 |
>  | Group 2 (126) | 0.86 | 0.91 | 0.88 | 0.84 | 0.04 |
>
> **(Q4)** Expand discussion of real-world feasibility.
>
> **Response:** We agree that one limitation of this study is the use of a synthetic dataset, which we selected to explicitly control the presence and magnitude of morphological bias. This controlled setting enables a clear link between sampling strategy and performance bias.
>
> *Evaluating* our framework in real-world data scenarios requires (i) identification of a dataset with known biases to confirm that a measurable disparity exists, and (ii) reliable reference segmentations for training, ideally generated without performance biases.
> In practice, finding such datasets is extremely challenging.
> Automatic segmentations may carry performance biases making fairness evaluation challenging, and manual annotations by experts are scarce. Moreover, real cohorts may exhibit subtle or region-specific morphometric differences (or none at all), and the existence of such differences cannot be assumed a priori.
> These challenges led us to perform these experiments solely with synthetic data. Our results support the use of a weighted sampling strategy to avoid performance bias associated with group level attributes.
> If one were to *translate* or apply our framework in a real-world scenario, one would need to identify one or more sensitive attributes based on apriori hypotheses (structure X has been reported to be larger in sub-population Y) and guide the sampling strategy using the weighted localized entropy (eq. 2 in the manuscript).

---

> > ### Comment · Reviewer_BUYe · 2026-02-02
> >
> > I would like to thank the authors for addressing my concerns. The clarity of the experiments has been significantly improved in the revised version, and the figures are easier to read. While I concur with Reviewer qMDW on the necessity of a real-world experiment, even with weak labels, I do believe that the methodological contribution to active learning with fairness constraints should clearly be presented and therefore recommand weak acceptance.

---

### Official Review · Reviewer_WQ2X · 2026-01-07

**Confidence:** 4
**Preliminary Rating:** 3
**Final Rating:** 4

**Summary:**

The authors propose a novel active learning sampling strategy based on group-weighted entropy to prioritize under-performing groups and therefore improve fairness metrics in brain segmentation. Using synthetic 3D brain MRI with controlled morphological biases, they show that this strategy reduces subgroup disparities when using active learning for segmenting a regional brain volume.

**Strengths:**

The paper is generally well written. The motivation and description for the proposed sampling strategy is clear and appears to be valuable, as the figures clearly convey improved fairness-weighted performance. Overall, the experiments are designed well (but need improvements in how they are communicated to the reader).

**Weaknesses:**

In my view, one of the main weaknesses in this paper comes from the data setup and how it is presented. I am familiar with SimBA, but the particular sampling strategy in this work is not fully clear – see question #1 below. My understanding is that the weak/strong bias comes from the global deformation, while group 1/group 2 comes from a localized deformation. However, it is not entirely clear what these different scenarios are meant to represent. Is it just that the strong bias and group 1 data are expected to be harder for a model to segment? Since this is a non-traditional data setup, it would be beneficial for the authors to explain this when introducing the data.

Furthermore, since the experimental setup is a bit complex (strong vs. weak bias, group 1 vs. group 2, 3 baselines, 3 AL scenarios), the paper can be hard to follow. This work would really benefit from a graphical abstract illustrating the experimental design.

The design of the figures also has significant room for improvement (see detailed comments).

**Detailed Comments:**

As mentioned above, it would help if the different data scenarios were explained in relation to the task, i.e. it could be made clearer that the groups correspond to localized deformation in the target segmentation region earlier in the paper, and how both the local and global deformations are expected to impact segmentation and AL. Along these lines, group 1/group 2 and strong/weak bias could have more informative names to make the paper easier to follow.

Table 1: Caption should specify that this is for baselines, and arrows for DSC/ESSP could be included to reiterate that higher ESSP is desirable.

Figure design: Rows and columns could be labelled to ease interpretation. The abrupt cropping of the right hand side of each graph makes the figures appear messy/rushed. The X axis could be cleaner/with less text (e.g. could just be labelled in terms of AL cycles). I would also recommend having the same Y axis scale for all plots of the same metric (i.e. all ESSP plots, all delta plots, all DSC plots) to make interpretation more straightforward.

**Justification Of Final Rating:**

This paper tackles a challenge at the intersection of active learning, fairness, and segmentation, which I think would be valuable to discuss and share with the community. As an initial exploration, I don't think the exclusive use of synthetic data is a drawback per se, but I agree that future work should validate this method on real-world datasets.

**Justification Of The Preliminary Rating:**

While the contribution of the active learning sampling strategy appears to be valuable, the experimental design should be explained more clearly (ideally with a graphical abstract-type figure) as it is currently hard to follow. My questions/concerns about the synthetic data setup are also crucial for understanding and improving the paper.

**Questions To Address In The Rebuttal:**

1. Is it correct that the strong vs. weak bias effects correspond to the global deformation? This is not fully clear since the paper mentions localized bias and disease effects just prior to describing the strong and weak scenarios. Also, the sampling strategy for the localized deformation is not provided - what was the deformation sampled from for the left caudate effect?

2. What is the purpose of the global deformation if the entropy is only computed over a masked region anyway? Why not just compare degrees of deformation in the left caudate specifically for a fixed amount of global variation?

3. What do the shaded regions and dashed lines represent on the graphs? Are these intervals available for the Table 1 baseline as well?

---

> ### Author Response · Authors · 2026-01-25
>
> **Q1** Is it correct that the strong vs. weak bias effects correspond to the global deformation..?
>
> **Response:** We apologize for the lack of clarity in the manuscript about the experimental design and synthetic dataset.
> Strong and weak biases do not correspond to global deformations. The global deformation represents a general inter-subject variability (ISV) component, which is sampled from a whole-brain PCA deformation model. This component is applied to all subjects to introduce realistic anatomical variability and is not tied to the bias condition.
> By contrast, the strong and weak bias correspond to the magnitude of a localized deformation applied to the target structure (left caudate). Concretely, the localized effects are sampled from ROI-specific PCA models.
> In our work, the Group1 images were generated using a global deformation combined with a localized deformations (the bias effect) of the left caudate. Images in Group2 are generated using only global deformations with no localized deformation.
> The original SimBA framework allows for the combination of one global deformation with two local deformations (disease and bias effects respectively).
> In our work, no disease effect is applied to the images in Group1 or Group2. We only apply a bias effect, as we want to test the behavior of AL sampling strategies in the most controlled scenario, where we have a population with two distinct groups differentiated by a different morphology in one specific ROI (the left caudate).
>
> **Global deformation.** To model inter-subject anatomical variability, we sample a global deformation coefficient (ISV) from a Gaussian distribution like $\mathcal{N}(0,1)$ subject to predefined bounds. This scalar parameter controls the magnitude of the global deformation field.
>
> **Localized (bias) deformation.** To generate a localized morphological effect in the left caudate, we sample a bias magnitude coefficient from a Gaussian distribution, and use it to scale the first principal component of the ROI-specific deformation model. In the strong-bias setting, the coefficient is drawn from $\mathcal{N}(4,2^{2})$, whereas in the weak-bias setting it is drawn from $\mathcal{N}(2,2^{2})$. This localized deformation is applied only to samples labeled as biased (Group1); non-biased samples (Group2) receive only the global deformation.
>
> **Updated text** "The global deformations can be used to mimic “regular” anatomical variation and localized deformations can be utilized to show localized “bias or disease” effects. The deformation(s) applied to each case is unique and controlled by sampling from a principal component (PC) representation of deformation fields. The amount of localized deformation used to have bias effect is varied by scaling the first component of the PC representation by a scalar sampled from $\mathcal{N}(\mu, \sigma)$."
>
> **Q2**  What is the purpose of the global deformation if the entropy is only computed over a masked region anyway...?
>
> **Response:** We will clarify that while our acquisition metric (entropy) is localized to filter out noise, the input data requires global deformation to model general inter-subject anatomical variability and ensure the segmentation task remains non-trivial. The global deformation simulates essential inter-subject variability (ISV). If we fixed the global variation as suggested, every synthetic image would share an identical background anatomy. The 3D U-Net could simply memorize the position of the left caudate relative to the fixed global structures, achieving artificially high performance.
>
> **Q3**  What do the shaded regions and dashed lines represent on the graphs..?
>
> **Response:** Because each active learning experiment is repeated 5 times, we wanted to display the variability of the various sampling strategies. In order to do so, we use the functional box plot, an informative exploratory tool for visualizing functional data.
> The functional box plots are constructed from the performance curves obtained from 5 experimental runs. The curves are ordered from the center outward using the notion of band depth, which enables the definition of functional quantiles. The shaded regions denote the 50\% central region. It provides a robust measure of spread of the central half of the curves. Outlying curves can be identified using an empirical rule based on 1.5 times the size of the 50\% central region. The dashed lines indicate the outer region, which serves as the functional analogue of the whiskers in a classical box plot. The median curve represents the most central observation.
>
> Regarding Table1: in the current submission, Table1 reports single baseline trainings for each cohort configuration (All / Group1 / Group2). Functional boxplots are not applicable in this case as there are no curves representing the AL sampling behavior.

---

> > ### Comment · Reviewer_WQ2X · 2026-01-27
> >
> > Thank you for the clarification.
> >
> > I think strong vs. weak and group 1 vs. group 2 is still a bit confusing if one were to just read the manuscript. In my view, this is because strong vs. weak terms are introduced before group 1 vs. group 2, even though strong vs. weak is essentially a subset of group 1. I think it would be much clearer to readers if Sec 3.1 was rearranged wrt this "hierarchy" of group terminology (i.e. introduce that group 1/group 2 corresponds to presence/absence of localized effects, and then explain how within group 1, there exists strong bias/weak bias).

---

> > > ### Author Response · Authors · 2026-01-30
> > >
> > > We agree and will revise Sec. 3.1 to follow the mentioned hierarchy: we will first define Group 1 vs. Group 2 as the presence vs. absence of the localized deformation, and then explain that strong vs. weak refers to the magnitude setting of the localized deformation applied exclusively within Group 1.

---

> > ### Comment · Reviewer_WQ2X · 2026-01-27
> >
> > I also think that it may be important for the authors to consider how the global deformations for each group could be impacting comparisons. Especially given the relatively small sample sizes, the actual distribution of sampled global variation in the strong vs. weak bias scenarios could be having an effect on the results (e.g. if Group 1 happened to correspond to images with a stronger magnitude of ISV effects in the weak bias scenario, compared to the strong bias scenario). Unless this sampling was done in a systematic “counterfactual” setup (as in 10.1016/j.ebiom.2024.105501), it may be relevant to include this as a potential limitation.

---

> > > ### Author Response · Authors · 2026-01-30
> > >
> > > We agree that, with finite samples, accidental differences in the distribution of global inter-subject variability (ISV) could confound strong vs. weak comparisons if not controlled. However, we try to minimize this problem by randomly sampling from a gaussian distribution. The global deformation magnitude is drawn independently from a bounded Gaussian distribution to model general anatomical variability, and the strong/weak condition is defined solely by the distribution of the localized deformation coefficient. This design reduces systematic coupling between bias strength and global ISV. Nonetheless, we acknowledge that random sampling cannot guarantee perfectly matched ISV distributions in every finite instantiation.

---

### Author Rebuttal · Authors · 2026-01-25

**Rebuttal:**

We thank the reviewers for their detailed reading of the manuscript and for their helpful comments. We have addressed the comments in the official answers and revised the manuscript accordingly. We changed the figures, added the requested tables, and expanded the discussion to address remaining ambiguities and improve clarity in the revised manuscript, which is attached as the supporting material.

The main points are: (i) clarified the experimental design and the distinction between global and the localized (bias) deformation; (ii) clarified the role of global deformation even when uncertainty is computed within an ROI mask; (iii) explained the functional box-plot elements; (iv) added tables summarizing experiment conditions and training-set compositions; (v) improved figure readability and expanded the discussion on real-world feasibility and limitations of evaluating fairness on real datasets; and (vi) expanded the discussion of the “Group 1-only” baseline and ESSP.

We hope these revisions resolve the concerns and strengthen the clarity and contribution of the submission.

**Supporting Material:**

/attachment/022509a5eada8b424f62a79fd4be44f843d8a4b2.pdf

---

### Comment · Area_Chair_zijy · 2026-01-29
**Discussion phase**

Dear Reviewers, the manuscript has now entered the discussion phase.

We kindly invite you to evaluate the authors’ responses and the revised manuscript, and to engage in discussion with the authors to address any remaining questions or unresolved points.

Once you have completed your review and discussion, submit your final rating. Please complete this step by selecting “Edit” → “Official Review” no later than February 1, 2026, at 23:59 AoE.

---

### Meta-Review · Area_Chair_zijy · 2026-02-07

**Recommendation:** Accept (Poster)
**Confidence:** 5

**Metareview:**

The paper is a clear acceptance. The authors introduce a novel active learning sampling strategy that uses group-weighted entropy to address underperforming groups and enhance fairness in brain segmentation. While reviewers suggested that validation on clinical data would be beneficial, reviewers agreed that the synthetic data experiments are sufficient for this study.

---

### Decision · Program_Chairs · 2026-02-13

Accept (Poster)